# ViTKD: Feature-based Knowledge Distillation for Vision Transformers

## Abstract

Knowledge Distillation (KD) has been extensively studied as a means to enhance the performance of smaller models in Convolutional Neural Networks (CNNs). Recently, the Vision Transformer (ViT) has demonstrated remarkable success in various computer vision tasks, leading to an increased demand for KD in ViT. However, while logit-based KD has been applied to ViT, other feature-based KD methods for CNNs cannot be directly implemented due to the significant structure gap. In this paper, we conduct an analysis of the properties of different feature layers in ViT to identify a method for feature-based ViT distillation. Our findings reveal that both shallow and deep layers in ViT are equally important for distillation and require distinct distillation strategies. Based on these guidelines, we propose our feature-based method ViTKD, which mimics the shallow layers and generates the deep layer in the teacher. ViTKD leads to consistent and significant improvements in the students. On ImageNet-1K, we achieve performance boosts of $1.64\%$ for DeiT-Tiny, $1.40\%$ for DeiT-Small, and $1.70\%$ for DeiT-Base. Downstream tasks also demonstrate the superiority of ViTKD. Additionally, ViTKD and logit-based KD are complementary and can be applied together directly, further enhancing the student's performance. Specifically, DeiT-Tiny, Small, and Base achieve accuracies of $77.78\%$, $83.59\%$, and $85.41\%$, respectively, using this combined approach.

## 1 Introduction

Knowledge Distillation (KD) (Hinton et al., 2015) utilizes the output of the teacher model as soft labels to supervise the student model, bringing the lightweight models impressive improvements without extra costs for inference. It has been consistently explored for Convolutional Neural Network (CNN) models and applied to many vision tasks successfully, including image classification (Chen et al., 2021; Lin et al., 2022; Yang et al., 2020; Zhao et al., 2022; Zhou et al., 2020), object detection (Cao et al., 2022; Li et al., 2022a; Wang et al., 2022; Yang et al., 2022b; Zheng et al., 2022), and semantic segmentation (He et al., 2019; Liu et al., 2019; Shu et al., 2021; Yang et al., 2022a).

Recently, Vision Transformer (ViT) (Dosovitskiy et al., 2021) has achieved great success in image classification and inspired various transformers (Han et al., 2021; Liu et al., 2021; Touvron et al., 2021b; Yuan et al., 2021). Similar to CNN models, the ViT models generally need more parameters to achieve better performance, making them harder to be deployed. Therefore, boosting the performance of small ViT models using KD is of great value. In this study, we explore *how to apply KD to ViT-based models.* One straightforward approach would be to transfer the KD methods used for CNNs to ViTs. In fact, some fundamental distillation works (Hinton et al., 2015; Romero et al., 2015) are structure-independent. For example, the logit-based distillation directly utilizes the model's final logit, enabling it to be used for both CNNs and ViTs. This has been confirmed by DeiT (Touvron et al., 2021a) and TinyViT (Wu et al., 2022).

However, most of the KD methods beyond logit-based distillation are specifically designed for CNN-based models and rely on intermediate features. Due to the vast architectural differences between CNNs and ViTs, these methods are not applicable to ViT-based models. While recent work MiniViT (Zhang et al., 2022) has employed self-attention distillation and hidden-state distillation for vision transformers with various stages, such as Swin-Transformer (Liu et al., 2021), it is still not viable for models with multiple encoder layers like ViT (Dosovitskiy et al., 2021).

Table 1: The distillation results of existing feature-based KD methods, including recent MGD and classical FitNet on the last layer, the last 6 layers, and all the 12 layers.

| Distillation setting | T: DeiT Small, S: DeiT Tiny | |
| --- | --- | --- |
| | Top-1 Acc. (%) | Top-5 Acc. (%) |
| *baseline* | *74.42* | *92.29* |
| MGD (Yang et al., 2022c) | 74.46 (+0.04) | 92.28(-0.01) |
| Last layer (FitNet (Romero et al., 2015)) | 73.36 (-1.06) | 91.88 (-0.41) |
| Last 6 layers (FitNet (Romero et al., 2015)) | 73.76 (-0.66) | 92.01 (-0.18) |
| All 12 layers (FitNet (Romero et al., 2015)) | 74.24 (-0.18) | 92.23 (-0.06) |

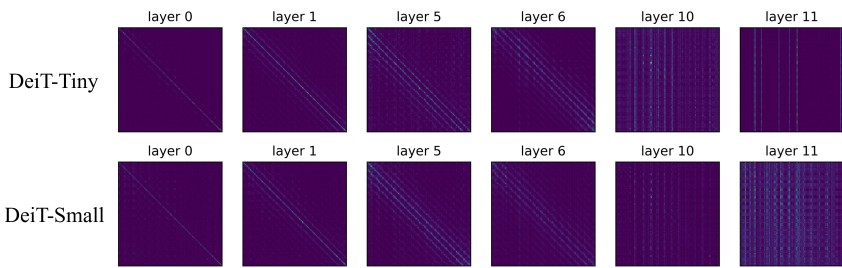

Figure 1: DeiT-Tiny's (upper) and DeiT-Small's (under) attention maps from shallow to deep layers. The X-axis and Y-axis mean the key and query tokens, respectively. The attention maps are obtained by *softmax* and reflects the response between the query and key tokens. The color is brighter with a larger response between the query and key tokens.

Before developing a new feature-based KD method for ViT, we first conduct simple studies with two structure-independent methods FitNet (Romero et al., 2015) and MGD (Yang et al., 2022c). We explored distilling knowledge from the last layer like CNN's general distillation, the last 6 layers like PKD (Sun et al., 2019) for BERT's (Devlin et al., 2019) distillation, and the whole 12 layers of a teacher model (DeiT-Small) to a student model (DeiT-Tiny). Surprisingly, the results for all the intuitive feature distillations shown in Tab. 1 are not satisfactory which consistently degrade the performance of the student (DeiT-Tiny). Specifically, the Top-1 accuracy of the student is just 73.36% when distilling on the last layer with FitNet. This distillation on the last layer is widely used for CNN's distillation, but here it causes a 1.06% accuracy drop. When distilling the whole 12 layers, the accuracy drop reduces to 0.18%. This preliminary study suggests that distillation on the shallow layers is as crucial as that on the deep layers in ViT's distillation.

To gain a better understanding of ViT's features, we visualize the attention maps of the student and teacher across various layers, as shown in Fig. 1. For the shallow layers (*e.g.*, layers 0 and 1), both the student and teacher mainly focus on the diagonal, indicating a self-attention pattern. In contrast, for the deeper layers (*e.g.*, layers 10 and 11), there is a greater difference between the attention patterns of the student and teacher. Attention is determined by a few sparse key tokens, and the student and teacher focus on different tokens. This discrepancy makes it challenging for the student to mimic the teacher's final feature directly. Therefore, our findings suggest that different layers may require different knowledge distillation methods.

Accordingly, we perform a series of controlled experiments to examine the effects of different distillation methods and different layers. As a consequence, we propose a nontrivial way for feature-based ViT distillation, named **ViTKD**. ViTKD treats the shallow and deep layers with different distillation methods, which is shown in Fig. 2. We conduct extensive experiments to demonstrate its effectiveness. For instance, we boost the student DeiT-Tiny from 74.42% to 76.06%, DeiT-Small from 80.55% to 81.95% and DeiT-Base from 81.76% to 83.46% on ImageNet-1K. Besides, all loss functions in ViTKD are only calculated on feature maps, so it can be easily combined with the logit-based distillation. With the combination, we can further advance the students' Top-1 accuracy to 77.78%, 83.59% and 85.41%. We also demonstrate the models trained with ViTKD are beneficial to downstream tasks like object detection, human pose estimation and semantic segmentation. In a nutshell, the contributions of this paper are:

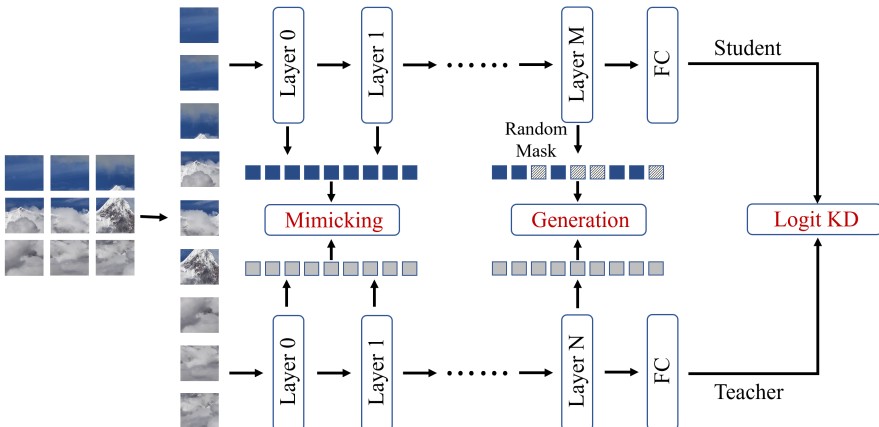

Figure 2: Illustration of the proposed ViTKD. ViTKD is a feature-based distillation method that includes shallow layers' *Mimicking* and deep layer's *Generation*. It can be directly combined with the output logit-based distillation method together.

- We reveal that the feature-based KD method for CNNs is unsuitable for ViTs. If we align them directly, it will result in a performance drop. Besides, the distillation on the shallow layers is also important for ViT, which differs from the conclusion in KD for CNNs.

- We provide an insight that treats different layers with different distillation methods. Based on this insight, we propose a simple and effective KD method named ViTKD.

- We verify the effectiveness of our method via extensive experiments on ImageNet (Deng et al., 2009), bringing significant performance gains. In addition, we also demonstrate the e superiority of the models trained with ViTKD for various downstream tasks on COCO (Lin et al., 2014) and ADE20K (Zhou et al., 2017).

## 2 RELATED WORK

### 2.1 VISION TRANSFORMER

Vision Transformer was proposed by (Dosovitskiy et al., 2021) for image classification. It applies the self-attention (Vaswani et al., 2017) architecture to computer vision tasks successfully. DeiT (Touvron et al., 2021a) explores the training setting and introduces logit-based distillation to ViT with another distillation token. CaiT (Touvron et al., 2021b) modifies the architecture and obtains a deeper model with vision transformer. T2T-ViT (Yuan et al., 2021) helps each token to get a better local feature. Swin Transformer (Liu et al., 2021) utilizes shifted windows to brings greater efficiency by limiting self-attention computation. Vision Transformer has also been applied to other downstream tasks, such as object detection (Li et al., 2022b; Liu et al., 2021) and semantic segmentation (Bao et al., 2022; Xie et al., 2021). However, such models consume many resources for inferring and need to be improved for better application.

### 2.2 KNOWLEDGE DISTILLATION

Knowledge distillation (KD) is a method to improve a compact model without extra time cost for inference. It was proposed by Hinton *et al.* (Hinton et al., 2015), which uses the teacher's output to guide the students. The following works can be divided into the logit-based methods and feature-based methods according to the distillation areas. WSLD (Zhou et al., 2020) analyzes soft labels from a perspective of bias-variance trade-off and distributes different weights for different samples. DKD (Zhao et al., 2022) decouple the logit according to the target class. SRRL (Yang et al., 2020) utilize the teacher's linear layer to help the student to get better features and logit. NKD (Yang et al., 2023) decomposes and normalizes classic KD, achieving state-of-the-art performance.

Feature-based KD methods calculate the distillation loss on the feature maps. FitNet (Romero et al., 2015) distills on the intermediate feature directly. RKD (Park et al., 2019) transfers the relation from the teacher's feature maps. CRD (Tian et al., 2019) introduces contrastive method for feature KD. KR (Chen et al., 2021) distills the knowledge from the teacher's multi-level features. MGD (Yang et al., 2022c) forces the student to generate the teacher's feature instead of mimicking. However, these feature-based methods are designed for CNNs.

## 3 METHODOLOGY

In this paper, we analyze the difference between ViT's different layers and treat them with different distillation methods. Specifically, for the shallow layers with a small difference, we force such layers to mimic the teacher's corresponding layers, learning how to focus on the tokens themselves. While for the deep layers with a big gap between student and teacher, which have stronger semantic information, we force the student to generate the teacher's feature instead of mimicking directly.

### 3.1 MIMICKING FOR SHALLOW LAYERS

As Fig. 1 shows, the student and teacher's shallow layers have similar attention. Besides, the attention appears mainly on the diagonal. So we force the student to mimic teacher's feature of the first two layers. The analysis of the choice for the shallow layers is in Sec. 5.2. For each sample, we can denote student's and teacher's feature as $\mathcal{F}^S \in \mathcal{R}^{N \times D_S}$ and $\mathcal{F}^T \in \mathcal{R}^{N \times D_T}$, respectively. For the *mimicking* method on shallow layers, we utilize a linear layer to align the embedding dimension of the student's $D_S$ and the teacher's $D_T$. The mimicking loss for shallow layers' distillation is as:

$$\mathcal{L}_{lr} = \sum_{i=1}^{N} \sum_{j=1}^{D_T} \left( \mathcal{F}_{i,j}^T - fc(\mathcal{F}^S)_{i,j} \right)^2, \tag{1}$$

where $fc(\cdot)$ is a linear layer to reshape the $\mathcal{F}^S$ to the same dimension as $\mathcal{F}^T$. $N, D_T$ denote the number of patch tokens and the embedding dimension of the teacher's feature.

### 3.2 GENERATION FOR DEEP LAYERS

For the deep layers, the student's and teacher's features become much more different and *mimicking* method fails on it, as shown in Tab. 1. So we try to utilize student's last feature to generate the teacher's last feature, avoiding mimicking directly. The analysis of the choice for the deep layers is in Sec. 5.2. The last feature has the best representation of the original input image. Such feature tokens already contain the information of adjacent tokens to a certain extent. Therefore, we can use partial tokens to generate the complete feature map. This way aims at generating the teacher's feature by student's masked feature, which can help the student achieve a better representation.

We first also use a linear layer to align the student's and teacher's feature embedding dimensions. Then, we set a random mask $Mask \in \mathcal{R}^{N \times 1}$ and use the learnable masked tokens to replace the student's original feature tokens:

$$\hat{\mathcal{F}}_i^S = \begin{cases} masked\ token, & \text{if } r_i < \lambda \\ original\ token, & \text{Otherwise,} \end{cases} \tag{2}$$

$$Mask_i = \begin{cases} 1, & \text{if } r_i < \lambda \\ 0, & \text{Otherwise,} \end{cases} \tag{3}$$

where $r_i$ is a random number uniformly distributed in $[0, 1]$ and $i \in [0, N-1]$ is the coordinates of the tokens dimension. $\lambda$ controls the masked ratio. The $masked\ token$ is the parameter to learn and will be updated during training.

Finally, we use the new masked feature $\hat{\mathcal{F}}_i^S$ to generate the teacher's full feature through a generative block $\mathcal{G}$, which can be formulated as follows:

$$\mathcal{G}(\hat{\mathcal{F}}^S) \longrightarrow \mathcal{F}^T. \tag{4}$$

For the generative block $\mathcal{G}$, we apply a convolutional projector, which includes two $3\times3$ conventional layers and one activation layer $ReLU$. Finally, we only calculate the loss of the masked tokens. For the *generation* method for deep layer, we design the distillation loss $\mathcal{L}_{gen}$ as:

$$\mathcal{L}_{gen} = \sum_{i=1}^{N} \sum_{j=1}^{D} Mask_i \big(\mathcal{F}_{i,j}^T - \mathcal{G}(\hat{\mathcal{F}}_{i,j}^S)\big)^2. \tag{5}$$

### 3.3 ViTKD

Combing the distillation on shallow (**first two**) and deep (**last**) layers, we propose ViTKD, as shown in Fig. 2. When the number of the student's and teacher's layers is different, we also pick the first two and last layers for distillation. To sum up, we train the student model with the total loss:

$$\mathcal{L} = \mathcal{L}_{ori} + \alpha\mathcal{L}_{lr} + \beta\mathcal{L}_{gen}, \tag{6}$$

where $\mathcal{L}_{ori}$ is the original loss for the models. $\alpha$ and $\beta$ are two hyper-parameters.

## 4 EXPERIMENT

### 4.1 SETTINGS

**Datasets.** We explore ViTKD on ImageNet-1K (Deng et al., 2009), which contains 1000 categories. We use 1.2 million images for training and 50k images to evaluate the performance. For downstream tasks, we evaluate our model on COCO (Lin et al., 2014) and ADE20K (Zhou et al., 2017).

**Implementation details.** ViTKD uses $\alpha$ and $\beta$ to balance the distillation loss in Eq. 6. Another hyper-parameter $\lambda$ is used to adjust the masked ratio for deep layer distillation in Eq. 2. We adopt the hyper-parameters $\{\alpha = 3 \times 10^{-5}, \beta = 3 \times 10^{-6}, \lambda = 0.5\}$ for all the experiments. Besides, to keep the model to be the same for the feature and logit distillation, we remove the extra distillation token which is used for logit distillation in DeiT. The image resolution for all the experiments is $224\times224$. The experiments are conducted on 8 Nvidia 3090 GPUs with MMClassification (Contributors, 2020a) in Pytorch (Paszke et al., 2019). Unless specified, we evaluate the model with the last epoch. The training setting follows DeiT (Touvron et al., 2021a), which is trained with Mixup, Cutmix, RandAugment and Erasing. We use Adamw as the optimizer with 1024 batch size. The learning rate with warm up decays as cosine. We take larger DeiT (Touvron et al., 2021a) and DeiT III (Touvron et al., 2022) models as the teacher to distill lighter DeiT models. The DeiT teacher is trained from scratch on ImageNet-1K, and DeiT III teacher is pre-trained on ImageNet-21K. More results and analyses are shown in Appendix. A.

### 4.2 MAIN RESULTS

Our ViTKD can bring ViT-liked models better performance via feature distillation. To prove this, we first conduct experiments with different teacher-student distillation pairs on ImageNet, as shown in Tab. 2. We compare our ViTKD with two other methods, including FitNet (Romero et al., 2015) and recent MGD (Yang et al., 2022c). To further show the effectiveness of our ViTKD, we also compare it with original KD (Hinton et al., 2015) and Hard KD (Touvron et al., 2021a) from DeiT. As shown in Tab. 2, ViTKD surpasses other feature-based methods significantly and brings remarkable accuracy gains, *e.g.*, the DeiT III-Small (Touvron et al., 2022) teacher boosts the student's Top-1 accuracy from 74.42% to 76.06%. Besides, ViTKD brings comparable improvements as DeiT's Hard KD and even surpasses the classic KD method. Comparing the results between different teachers, we find the student achieves better performance with a stronger teacher, *e.g.*, the student DeiT-Tiny achieves 75.40% and 76.06% Top-1 accuracy with the DeiT-Small and DeiT III-Small teacher, respectively.

We also conduct experiments with more and larger models to show the generalization of ViTKD, as shown in Tab. 3. For larger models such as Swin-T (Liu et al., 2021), DeiT-S and DeiT-B, ViTKD can also bring them significant improvements, helping it to achieve 81.70%, 81.95% and 83.46%, respectively. Besides, ViTKD is a feature-based method and can be combined with other logit-based methods to further improve the student. This is also one of the feature-based methods' advantages. Therefore, we try to add the state-of-the-art logit-based method NKD (Yang et al., 2023) to ViTKD,

Table 2: We reproduce the results of DeiT with MMClassification on ImageNet-1K. ∗ indicates the teacher is pre-trained on ImageNet-21K. We compare with two other feature-based distillation methods that can be applied to ViTs.

| Teacher | Type | Student | Top-1 Accuracy | Top-5 Accuracy |
|---------|------|---------|----------------|----------------|
| DeiT-Small (80.69) | baseline | DeiT-Tiny | 74.42 | 92.29 |
| | logit | KD | 75.01 (+0.59) | 92.52 |
| | | Hard KD | **75.10** (+0.68) | **92.55** |
| | feature | FitNet | 73.36 (-1.06) | 91.88 |
| | | MGD | 74.46 (+0.04) | 92.28 |
| | | **ViTKD (Ours)** | **75.40** (+0.98) | **92.66** |
| DeiT III-Small∗ (82.76) | baseline | DeiT-Tiny | 74.42 | 92.29 |
| | logit | KD | 76.01 (+1.59) | 93.26 |
| | | Hard KD | **76.08** (+1.66) | **93.30** |
| | feature | FitNet | 73.72 (-0.70) | 91.88 |
| | | MGD | 74.53 (+0.07) | 92.28 |
| | | **ViTKD (Ours)** | **76.06** (+1.64) | **93.16** |

Table 3: Results of distilling more and larger models on ImageNet-1K. ∗ indicates the teacher is pre-trained on ImageNet-21K.

| Teacher | Student | Baseline | + ViTKD |
|---------|---------|----------|---------|
| Swin-B∗ | Swin-T | 81.18 | 81.70 (+0.52) |
| DeiT III-B∗ | DeiT-S | 80.55 | 81.95 (+1.40) |
| DeiT III-L∗ | DeiT-B | 81.76 | 83.46 (+1.70) |

Table 4: The downstream tasks results with ViTKD.

| Dataset | Metric | DeiT-S | + ViTKD |
|---------|--------|--------|---------|
| ImageNet | Top-1 Acc. | 80.55 | 81.95 (+1.40) |
| COCO | $AP^{box}$ | 45.07 | 46.28 (+1.21) |
| | $AP^{mask}$ | 40.14 | 41.05 (+0.91) |
| | $AP^{pose}$ | 71.80 | 72.80 (+1.00) |
| ADE20K | mIOU | 42.96 | 44.94 (+1.98) |

as shown in Fig. 3. In this way, different students all get another significant accuracy improvement, *e.g.*, the student DeiT-Small gets another 1.64% gains and achieves 83.59% Top-1 accuracy with a DeiT III-Base teacher. Surprisingly, the student DeiT-Small is just trained on ImageNet-1K, but its performance surpasses DeiT III-Small's 82.76%, which needs to be pre-trained on ImageNet-21K.

### 4.3 DOWNSTREAM TASKS

The model trained with ViTKD achieves significant improvements for classification. To further evaluate the effectiveness of the model with ViTKD, we try to apply it to various downstream tasks. For object detection on COCO (Lin et al., 2014), we use Mask-RCNN (He et al., 2017) as the detector and follow the setting from ViT-Det (Li et al., 2022b) on detectron2 (Wu et al., 2019). For pose estimation, we use ViTPose (Xu et al., 2022) for the 17 body key points. For segmentation on ADE20K (Zhou et al., 2017), we use UPerNet (Xiao et al., 2018) and train it on MMSegmentation (Contributors, 2020b).

As the results in Tab. 4, DeiT-Small trained with ViTKD brings the detector 1.21 Box mAP and 0.91 Mask mAP gains. For human pose estimation, ViTKD brings 1.00 AP gains. For segmentation, ViTKD boosts the mIoU performance from 42.96 to 44.94. The results demonstrate the model trained with ViTKD has not only better performance for classification but also stronger semantic information for various downstream tasks.

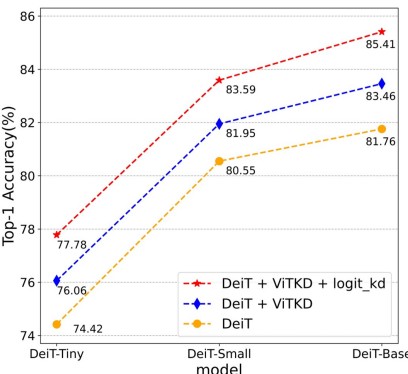

Figure 3: Results of combing ViTKD with logit-based KD on ImageNet-1K. The teacher for DeiT-T, S and B is DeiT III-S, B, and L, respectively. The teachers are pre-trained on ImageNet-21K.

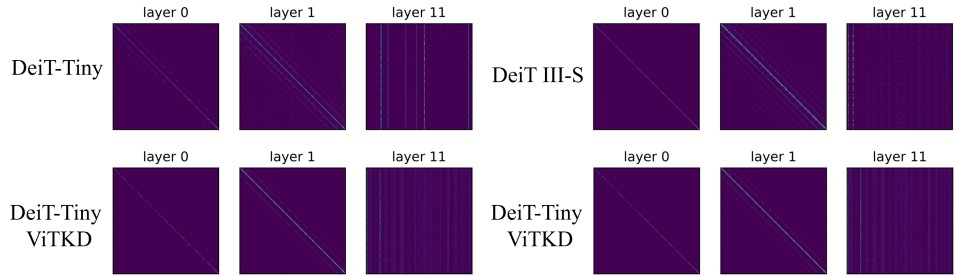

Figure 4: Visualization of the average attention map from the DeiT-Tiny (student), DeiT III-Small (teacher) and DeiT-Tiny after distillation with ViTKD. For a convenient comparison, the two visualizations in the second row are both DeiT-Tiny with ViTKD.

Table 5: The effects of distillation on different layers.

| | Layers | Top-1 Acc. (%) |
|---|---|---|
| shallow | 0 | 75.01 |
| | 0,1 | 75.12 |
| | 5,6 | 75.01 |
| | 0,1,6 | 75.14 |
| | 0,1,2,3,4,5,6 | 75.16 |
| deep | 11 | 74.72 |
| | 10,11 | 74.70 |

Table 6: Ablation study results of the losses of shallow layers' *Mimicking* and deep layer's *Generation*. ∗ means the teacher is pre-trained on ImageNet-21K.

| Losses | DeiT-Tiny (Student) | | | |
|---|---|---|---|---|
| $\mathcal{L}_{lr}$ | - | ✓ | - | ✓ |
| $\mathcal{L}_{gen}$ | - | - | ✓ | ✓ |
| DeiT-S (T) | 74.42 | 75.12 | 74.72 | **75.40** |
| DeiT III-S* (T) | 74.42 | 75.31 | 74.79 | **76.06** |

## 5 ANALYSES

### 5.1 STUDENT'S ATTENTION AFTER DISTILLATION

ViTKD combines mimicking and generation methods for different layers for ViT-based models. In this subsection, we present a visualization and comparison of the average attention maps from the teacher, student, and student with ViTKD to explore how ViTKD influences the student, as shown in Fig. 4. Comparing the attention maps between the original student and teacher, we observe a significant and much smaller difference in the deep and shallow layers, respectively. After distillation with ViTKD, the student's shallow layers have attention maps similar to those of the teacher. However, for the deep layer, as described in Sec.1, the significant gap between the attention maps of the teacher and student makes it difficult for the student to mimic the teacher directly. ViTKD forces the learnable masked tokens to be similar to those of the teacher, instead of using the original tokens of the student. As a result, after our ViTKD's generation, the student's deep features are still dissimilar to those of the teacher. However, ViTKD helps the student generate the teacher's features with its random masked tokens, resulting in the student's deep layer having stronger semantic information. We demonstrate this by using the deep layer for downstream tasks in Sec. 4.3, including detection, human pose estimation, and segmentation.

### 5.2 DIFFERENT LAYERS FOR DISTILLATION

As shown in Fig. 1, the attention distributions of the middle layers are also similar to that of shallow layers. Accordingly, we first explore the effects of distillation on such layers in Tab. 5. In general, distillation on either the shallow or middle layers can benefit the student. Besides comparing the improvements from different layers, we find that the knowledge from the shallow layers is much more helpful than that from the middle layer for distillation. Furthermore, when distilling the whole first seven layers together, the accuracy improvement is just $0.04\%$ above the first two layers. Considering the trade-offs between time consumption and performance, we choose the first two layers for shallow distillation eventually. While for the deep layer's distillation, calculating distillation loss on the last layer takes less time but brings more improvement.

Table 7: The comparisons of different generative blocks for deep layer's distillation on ImageNet-1K. ∗ indicates the model is pre-trained on ImageNet-21K.

| | Teacher | DeiT-Small (80.69) | DeiT III-Small∗ (82.76) |
|---|---|---|---|
| | Student | DeiT-Tiny (74.42) | DeiT-Tiny (74.42) |
| *Generation* | Cross-attention | 73.77 (-0.65) | 73.98 (-0.44) |
| | Self-attention | 74.61 (+0.19) | 74.65 (+0.23) |
| | Conv. projector | **74.72** (+0.30) | **74.79** (+0.37) |

Table 8: The comparisons of different alignments for shallow layers' distillation on ImageNet-1K.

| | Teacher | DeiT III-S∗ (82.76) | DeiT-B (81.76) |
|---|---|---|---|
| | Student | DeiT-T (74.42) | DeiT-T (74.42) |
| *Mimicking* | Linear layer | **75.31** (+0.89) | **75.15** (+0.73) |
| | Correlation matrix | 74.94 (+0.52) | 75.01 (+0.59) |

### 5.3 EFFECTS OF SHALLOW AND DEEP LAYERS' LOSSES

As described in the method, we distill the shallow layers and deep layers by mimicking and generation, respectively. In this subsection, we conduct experiments of *Mimicking* loss $\mathcal{L}_{lr}$ and *Generation* loss $\mathcal{L}_{gen}$ to investigate their influences on the student with DeiT-Tiny. As shown in Tab. 6, both the knowledge from shallow and deep layers are helpful for the student. When just applying a single loss, the $\mathcal{L}_{lr}$ on shallow layers benefits the student much more than $\mathcal{L}_{gen}$ on the deep layer. This phenomenon shows that incipient attention knowledge really matters for ViT's feature distillation, which is completely different from the CNN-based model's feature distillation. Furthermore, these two losses are complementary to each other. For example, when combing $\mathcal{L}_{lr}$ and $\mathcal{L}_{gen}$ together, the student with a DeiT III-Small teacher achieve 76.06% Top-1 Accuracy, which is much higher than just applying $\mathcal{L}_{lr}$'s 75.31% and $\mathcal{L}_{gen}$'s 74.79%.

### 5.4 DIFFERENT GENERATIVE BLOCK FOR DEEP LAYER'S DISTILLATION

For *generation*, we randomly mask the student's tokens and utilize a generative block to restore the feature. In this section, we discuss the effects of different generative blocks, including cross-attention block (Chen et al., 2022b), self-attention block (He et al., 2022), and convolutional projector (Yang et al., 2022c). We use two teachers to distill the student DeiT-Tiny on ImageNet-1K. As our results shown in Tab. 7, the cross-attention blocks impair the student's performance noticeably. Instead, the self-attention and convolutional block can improve the accuracy of the student. The largest gains are obtained by using the convolutional projector as the generative block.

### 5.5 DIFFERENT ALIGNMENTS FOR SHALLOW LAYERS' DISTILLATION

When using *mimicking*, we align the embedding dimensions of the student and the teacher by a linear layer. Here we try another alignment way called correlation matrix to describe the response among different tokens and force the student to learn the correlation matrix of the teacher's features. In this case, we do not need the adaption layer to align the embedding dimension. The correlation matrix can be calculated as:

$$\mathcal{M} = \frac{\mathcal{F}\mathcal{F}^{Tr}}{\sqrt{D}}, \tag{7}$$

where $\mathcal{F} \in \mathcal{R}^{N \times D}$ denotes the student or teacher's feature. $D$ is their embedding dimension and $Tr$ denotes transposition, so $\mathcal{F}^{Tr} \in \mathcal{R}^{D \times N}$. In this case, the student's and teacher's relation matrices have the same shape $\mathcal{M} \in \mathcal{R}^{N \times N}$ and describe the response between different tokens.

We pick the first two layers for distillation by mimicking in Tab. 8. Both alignments for transferring the knowledge from the shallow layer by directly mimicking make great progress. Mimicking by 'linear layer' performs better than the 'correlation matrix' way. When the teacher performs better, the 'linear layer' way benefits the student much more than the 'correlation matrix' way.

Table 9: Different modules for shallow layers.

Table 10: Sensitivity study of hyper-parameter $\lambda$, the masked ratio in deep layer's distillation.

| Teacher | DeiT-S (80.69) | DeiT III-S (82.76) |
|---------|----------------|--------------------|
| MHA | 75.06 | 75.02 |
| FFN | **75.12** | **75.31** |

| $\lambda$ | baseline | 0 | 0.3 | 0.5 | 0.7 | 0.9 |
|-----------|----------|-------|-------|-------|-------|-------|
| Acc. | 74.42 | 75.77 | 75.86 | 76.06 | 75.94 | 75.83 |

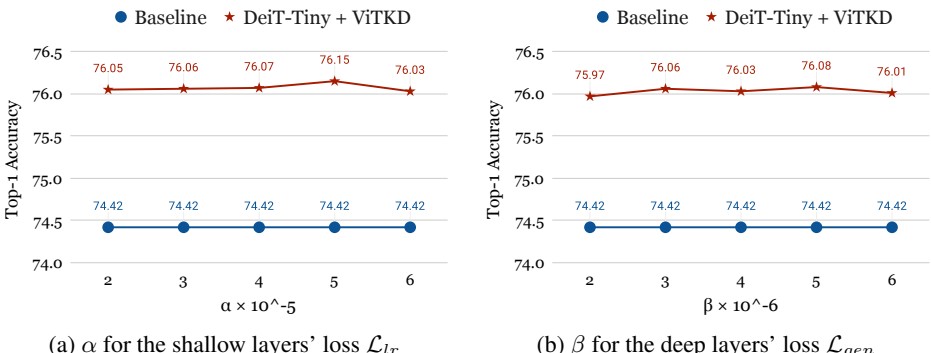

(a) $\alpha$ for the shallow layers' loss $\mathcal{L}_{lr}$    (b) $\beta$ for the deep layers' loss $\mathcal{L}_{gen}$

Figure 5: The sensitivity study of hyper-parameters $\alpha$ (a) and $\beta$ (b).

## 5.6 DIFFERENT MODULES FOR DISTILLATION

ViTs are built by stacking several encoder layers. Each encoder layer consists of a multi-head attention (MHA) module and a feed-forward network (FFN) module. We further conduct experiments on the shallow layers of the student DeiT-tiny to explore how to choose the modules for distillation. The results in Tab. 9 demonstrate that distilling on the MHA-out or FFN-out feature both benefit the student and the knowledge from FFN-out feature is better than that from MHA-out feature.

## 5.7 SENSITIVITY STUDY OF HYPER-PARAMETERS

In this paper, we use two hyper-parameters $\alpha$ and $\beta$ in Eq. 6 to balance the shallow layer's distillation loss $\mathcal{L}_{lr}$ and the deep layer's distillation loss $\mathcal{L}_{gen}$, respectively. To explore the sensitivity of the hyper-parameters, we conduct experiments by adopting DeiT III-Small to distill DeiT-Tiny on ImageNet-1K. As shown in Fig. 5a and Fig. 5b, ViTKD is not sensitive to $\alpha$ or $\beta$ which is just used for balancing the distillation loss. Specifically, when $\alpha$ varies from 2 to 6, the student's worst accuracy is 76.03%, which is just 0.12% lower than the highest accuracy. Besides, it is still 1.61% higher than the baseline model, demonstrating our ViTKD is not sensitive to the hyper-parameters for loss scale. As for the masked ratio $\lambda$ in Eq. 2, it is used to control the ratio of masked tokens. We conduct experiments to explore it, as shown in Tab. 10. When $\lambda$ is 0, it means there are no masked tokens and the performance is poor. When it is too large, e.g., 0.9, the left tokens are too poor to generate the teacher's feature and the performance is also affected. Besides, $\lambda$ is applied just for deep distillation. So the influence of ViTKD is limited. The lowest performance is still 1.35% higher than the baseline.

## 6 CONCLUSION

In this paper, we propose a feature-based distillation method for ViT-based models. We point out that distillation on the shallow layers is also important for ViTs and different layers need different distillation methods. Based on this insight, we propose a simple and effective method ViTKD, which includes the distillation on shallow layers via *mimicking* and deep layers via *generation*. ViTKD brings the student significant improvements on the image classification task and also benefits the downstream tasks. Besides, ViTKD is truly a feature-based method and can be easily combined with other logit-based methods to further improve the student's performance. ViTKD attempts to apply feature-based KD for ViTs. We believe the insight that treating different layers with different methods can still be further explored and be extended to other tasks.

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

Table 11: The distillation results of applying different methods to different layers.

| Methods | Layers | Top-1 Acc. (%) |
|---|---|---|
| *baseline* | - | *74.42* |
| Mimicking | shallow | 75.31 |
| | deep | 73.72 |
| | shallow + deep | 74.65 |
| Generation | shallow | 74.69 |
| | deep | 74.79 |
| | shallow + deep | 75.02 |
| ViTKD | shallow + deep | 76.06 |

Table 12: The comparisons of more methods on ImageNet-1k. The student model is DeiT-Small.

| Methods | Type | Top-1 Acc. (%) |
|---|---|---|
| DearKD (Chen et al., 2022a) | feature + logit | 81.5 |
| Manifold (Hao et al., 2022) | feature + logit | 82.2 |
| ViTKD | feature | 82.0 |
| ViTKD + logit-kd | feature + logit | 83.6 |

# A APPENDIX

## A.1 DIFFERENT METHODS FOR DIFFERENT LAYERS

Our ViTKD applies mimicking way to shallow layers and generation way to deep layer, respectively. Here we try to apply the mimicking or generation way to the shallow and deep layers together. As shown in Tab. 11, both mimicking and generation ways bring the student limited improvements. Our ViTKD achieves significant results and boosts the student from 74.42% to 76.06%.

## A.2 COMPARISON WITH MORE METHODS

We also compare our ViTKD with more methods which include both feature and logits distillation for ViT-liked models, as shown in Tab. 12. Our ViTKD achieves comparable results via feature distillation. When combining the logit-based kd together, the performance gets another significant improvement. The results demonstrate our ViTKD surpasses other methods in a fair setting.

## A.3 DOWNSTREAM TASKS OF LARGER MODELS

In Sec. 4.3, we report the downstream results of DeiT-Small. Here we add the object detection results of a larger model DeiT-Base with our distillation method, as shown in Tab. 13. The student also achieves consistent improvements and can be further improved with logit-based KD together.

## A.4 TIME COST OF VITKD

We report the time for a training epoch on 8 Nvidia 3090 GPUs in Tab. 14. We test DeiT-Tiny with a DeiT-Small teacher. The logit-based method also needs to obtain intermediate features before final logits. Therefore, it's reasonable to utilize features and logits together for distillation. For both settings, ViTKD's time consumption is similar to the logit-based method.

Table 13: The downstream tasks of larger models.

| Methods | Model | Type | COCO Det mAP |
|---|---|---|---|
| *baseline* | *DeiT-B* | - | *47.23* |
| ViTKD | DeiT-B | feature | 48.13 |
| ViTKD + logit-kd | DeiT-B | feature + logit | 48.83 |

Table 14: The downstream tasks of larger models.

| Methods | Type | Time (min) | Top-1 Acc. (%) |
|---|---|---|---|
| NKD | logit | 7.6 | 75.48 |
| ViTKD | feature | 7.8 | 75.40 |
| ViTKD + NKD | feature + logit | 7.9 | 76.18 |

Table 15: The average results of ViTKD. The teacher is DeiT-Small, and the student is DeiT-Tiny.

| Times | 1 | 2 | 3 | Average | Standard Deviation |
|---|---|---|---|---|---|
| Top-1 Acc. (%) | 76.06 | 76.10 | 76.08 | 76.08 | 0.016 |

## A.5 STANDARD DEVIATION OF THE EXPERIMENTS

As shown in Fig. 5, we take the sensitivity study of two hyper-parameters. Among the experiments, the best performance is 76.15%, and the worst is 75.97%, which is still 1.55% higher than the baseline. Here we run our experiment another two times to show the error bars in Tab. 15.

