# OpenReview forum: "ViTKD: Feature-based Knowledge Distillation for Vision Transformers"
_ICLR.cc/2024/Conference — ICLR 2024 Conference Withdrawn Submission_

### Official Review · Reviewer_1utk · 2023-10-28

**Soundness:** 2 fair
**Presentation:** 2 fair
**Contribution:** 2 fair
**Rating:** 3
**Confidence:** 5

**Summary:**

This paper studies the feasibility of applying CNN's feature distillation to the ViT model. The main contributions are two folds: 1) They found that the shallow layer distillation is important and they apply the traditional feature mimicking on the shallow layers. 2) They found that the feature mimicking produces sub-optimal results when distilling on the deep layers. The paper borrows the MGD method for deep-layer distillation. Experiments are sufficient and the results show reasonable improvement.

**Strengths:**

1. The experiments are sufficient to verify the effectiveness of the methods.

2. Revealing the differences between ViT distillation and CNN distillation is meaningful.

**Weaknesses:**

1. The technical contribution/novelty is quite limited. Firstly, applying feature-based distillation on the shallow layers is not novel since this is common in CNN distillation, e.g., FSP [1], AT [2], OFD [3], and VID [4]. One can easily apply them on all the layers. Second, the main discovery is the performance degradation when distilling the deep layers. However, the proposed method, ViTKD, mainly adopts MGD [5] as the solution to this problem, which is a direct application of CNN's distillation. If we look at Sec. 3.1 and 3.2, there is no fundamental difference between the listed equations and the ones in Sec.3 of the MGD paper. This makes the paper more like a technical report rather than a scientific paper.

2. There are inappropriate statements in the contributions listed at the end of the Introduction section. 1) "We reveal that the feature-based KD method for CNNs is unsuitable for ViTs." However, the proposed methods (Sec. 3.1, 3.2) are all CNN's distillation methods. 2) "The distillation on the shallow layers is also important for ViT, **which differs from the conclusion in KD for CNNs.**" Can you provide some references that explicitly make that conclusion on CNN KD? Is there any evidence before claiming that shallow layer distillation is not important for CNN KD?

3. The necessity of using random masks is low. As shown in Table 10, when $\lambda=0$, the method boosts the performance by +1.35 Acc. However, it can be further improved by only +0.29 Acc. if we carefully tune $\lambda$. This significantly undermines the motivation of deep layer’s Generation distillation, because one can see that the main improvement comes from the feature-based distillation just with one more additional block to process the student's features.


[1] Yim J, Joo D, Bae J, et al. A gift from knowledge distillation: Fast optimization, network minimization and transfer learning. CVPR 2017.

[2] Zagoruyko S, Komodakis N. Paying more attention to attention: Improving the performance of convolutional neural networks via attention transfer. ICLR 2017.

[3] Heo B, Kim J, Yun S, et al. A comprehensive overhaul of feature distillation. ICCV 2019.

[4] Ahn S, Hu S X, Damianou A, et al. Variational information distillation for knowledge transfer. CVPR 2019.

[5] Zhendong Yang, Zhe Li, Mingqi Shao, Dachuan Shi, Zehuan Yuan, and Chun Yuan. Masked generative distillation. ECCV 2022.

**Questions:**

see weaknesses

---

### Official Review · Reviewer_bvvH · 2023-10-29

**Soundness:** 3 good
**Presentation:** 3 good
**Contribution:** 3 good
**Rating:** 5
**Confidence:** 4

**Summary:**

This paper introduces ViTKD, a feature-based knowledge distillation method for vision transformers. The key idea is to apply different distillation methods to different layers of ViTs, such as mimicking for shallow layers and generation for deep layers. The authors analyze the characteristics of different feature layers in ViTs to illustrate insights and conduct comprehensive experiments on ImageNet-1K and various downstream tasks to show the effectiveness of ViTKD.

**Strengths:**

# Strengths
1.	The paper provides a novel insight that different layers of ViTs require different distillation methods, and proposes a simple and effective method based on this insight. The paper also analyzes the properties of different feature layers and attention maps of ViTs, which is helpful for understanding the behavior of ViTs.
2.	Extensive experiments are reported applying ViTKD to various vision transformers on ImageNet classification, showing consistent improvements.
3.	The paper is generally well-written and organized, making it easy to follow the proposed method and experimental results.

**Weaknesses:**

# Weakness
1.	Downstream results are only compared to the baseline, not other distilled or pruned small ViTs.
2.	The combination with logit-based distillation methods such as NKD is mentioned without sufficient explanation. It remains unclear whether more hyperparameters will be introduced on the basis of many hyperparameters (α, β, λ), or make the method more complicated for other reasons.
3.	In the main text, the author only compares the proposed method with a few baselines, and the logits-based methods compared are all from papers before 2021. The relatively new articles in Appendix A.2 are easily overlooked, so the comparison results of this part can be mentioned in the main text.
4.	More analysis could be provided to better understand how and why ViTKD works.  For example, in Figure 4 the authors could add attention maps of some middle layers (such as layers 5 and 6 aligned with Figure 1) to show how they change after distillation.
5.	The paper would benefit from a more detailed discussion of the limitations of the proposed approach. Providing insights into the potential weaknesses of the method would strengthen the paper's contribution.

**Questions:**

Please refer to the weaknesses mentioned above.

---

### Official Review · Reviewer_LtTD · 2023-10-30

**Soundness:** 2 fair
**Presentation:** 3 good
**Contribution:** 2 fair
**Rating:** 3
**Confidence:** 4

**Summary:**

This paper proposes ViTKD, a knowledge distllation method tailored for vision transformer. The authors claim that traditional feature distillation for CNN is not suitable for ViT. Thus, they propose feature generation using masked modeling. The method is validated in different backbones and downstream tasks.

**Strengths:**

1. Paper is well-written and easy to follow.
2. The authors claim SOTA performance in multiple tasks, including image classification, semantic segmentation and object detection.
3. The method is validated on multiple backbones such as DeiT and Swin-T.

**Weaknesses:**

1. The technical contribution is limited. $L_{lr}$  is proposed by FitNet [1].  $L_{gen}$ is borrowed from masked language (image ) modeling. The authors claim that they use NKD as the logit-based distillation method. This make the novelty weak.
2. The motivation is not clear. The paper do not explain why feature generation (i.e. masked modeling) is effective to vit-based architecture. If  Table 1 and Table 5 are the support for they method, then the motivation is empirical and not generalized. When it comes to different teacher-student pair, the motivation may not be met. For instance, in Table 3, the performance on swin transformer (+0.52) is limited compared to DeiT (>=1.4).
3. In 5.2, the authors use first two layers for feature disillation and the last layer for feature geneartion. However, they analysis do not provide enough insight to explain such decision. Hence, the experimental result looks random. When the student network architecture is changed, the same recipe can not guarantee the performance. In contrast, logit-based distillation is model-agnostic and general.
4. By my understanding, the result in Table 2 uses the technique of logit-based method NKD. The ablation study in Table 6 does not show the performance of ViTKD without such a loss, making the claimed performance unclear.
5. Regarding the transformer and knowledge distillation, this work misses some releated work such as DistillBert.

[1] Fitnets: Hints for thin deep nets. ICLR 2015

[2]DistilBERT, a distilled version of BERT: smaller, faster, cheaper and lighter

**Questions:**

1. If I do not misunderstand the description in 4.2, I am curious about the performance (Table 2) without a logit-based method. This result can help us to clear the contribution of this work.
2. What is the metric to identify the layers for either feature distillation or feature generation?
3. Table 4 shows improvement on different downstream tasks. However, ViTKD does not compare with other methods, making the claimed performance improvement unclear.

Given the concerns to the novelty and performance, I would suggest the author discussing these issues.

---

### Official Review · Reviewer_HQ1H · 2023-10-31

**Soundness:** 3 good
**Presentation:** 3 good
**Contribution:** 3 good
**Rating:** 6
**Confidence:** 4

**Summary:**

The paper tackles the problem of how to perform feature distillation with vision transformers (ViTs). Preliminary observation indicates that, the methods designed for CNN’s feature distillation are not applicable. Also, contrary to CNN’s feature distillation, in ViT’s the distillation for shallow layers is also important. For shallow layers the attention pattern between teacher and student are similar, however, this is not the case for deeper layers. Based on these observations, the paper proposes to handle the feature distillation for student’s shallow layers and deep layers differently. For student’s shallow layers, the goal is to mimick the teacher’s shallow layers whereas for student’s deeper layer, the goal is to generate the teachers deep feature after masking student’s deep feature. Experiments have been performed with DeiT models, where a larger capacity DeIT model acts as a teacher and a relatively smaller capacity model becomes the student. Results on different tasks, including image classification, and downstream tasks, claim to achieve superior performance than baselines.

**Strengths:**

- Developing effective feature distillation methods for ViTs is an important research direction because the ViTs are increasingly used in several computer vision tasks and their real-world deployment requires fast inference times.

- The idea of micmicking for student’s shallow layers and generating for student’s deeper layers is based on empirical observation that shallow layer features from student and teacher are similar in pattern while this is not the case with the deeper layers, which contain (strong) semantic information.

- Beyond image classification, experiments have been performed for downstream tasks, including object detection and semantic segmentation. Results claim to consistently surpass the baseline methods.

- The paper is mostly well-written and easier to go through. Ablation studies and some analyses have been reported on different design choices and hyperparameter sensitivity.

**Weaknesses:**

- The intuition behind why the generation mechanim was chosen to fill the gap between student’s deep layers features and teacher’s deep layer features is not very clear.

- The initial empirical analyses is just based on visualization and perhaps comparison based on quantification of the disparity between two corresponding set of features could be more convincing.

- What happens to student’s deep layer features after applying the method of ViTKD is not clearly discussed?

- How does the generative process between student’s deep feature and teacher’s deep feature allows student’s deep features to obtain better semantic information?

- Since there is a gap between the student’s deep feature layers and teacher’s deep feature, instead of resorting to generative way, wouldn’t some relaxed loss formulation give similar or even better performance?

**Questions:**

- Is it possible to use the proposed method for distillation with a teacher CNN model and a student ViT model?

- Is there any specific reason on why the cross attention as generative block provides relatively poor performance compared to conventional. projector in Tab. 7?

**Details Of Ethics Concerns:**

No ethics concerns.